# Screening of New Industrially Important Bacterial Strains for 1,3-Propanediol, 2,3-Butanediol and Ethanol Production through Biodiesel-Derived Glycerol Fermentations

**DOI:** 10.3390/microorganisms11061424

**Published:** 2023-05-28

**Authors:** Dimitris Karayannis, Gabriel Vasilakis, Ioannis Charisteidis, Alexandros Litinas, Eugenia Manolopoulou, Effie Tsakalidou, Seraphim Papanikolaou

**Affiliations:** 1Department of Food Science and Human Nutrition, Agricultural University of Athens, 75 Iera Odos, 11855 Athens, Greece; dimika96@icloud.com (D.K.); vasilakis@aua.gr (G.V.); mae@aua.gr (E.M.); et@aua.gr (E.T.); 2Verd S.A., 2nd Industrial Area of Volos, 37500 Velestino, Greece; gcharisteidis@verd.gr (I.C.); alitinas@verd.gr (A.L.)

**Keywords:** industrial biotechnology, biochemicals, *Hafnia alvei*, *Klebsiella oxytoca*, *Enterobacter ludwigii*

## Abstract

A study on the ability of new microbial strains to assimilate biodiesel-derived glycerol at low purity (75% *w*/*w*) and produce extra-cellular platform chemical compounds of major interest was carried out. After screening several bacterial strains under different fermentation conditions (e.g., pH, O_2_ availability, glycerol purity), three of the screened strains stood out for their high potential to produce valued-added products such as 2,3-butanediol (BDO), 1,3-propanediol (PDO) and ethanol (EtOH). The results indicate that under aerobic conditions, *Klebsiella oxytoca* ACA-DC 1581 produced BDO in high yield (Y_BDO/Gly_ = 0.46 g/g, corresponding to 94% of the maximum theoretical yield; Y_mt_) and titer, while under anaerobic conditions, *Citrobacter freundii* NRRL-B 2645 and *Enterobacter ludwigii* FMCC-204 produced PDO (Y_PDO/Gly_ = 0.56 g/g, 93% of Y_mt_) and EtOH (Y_EtOH/Gly_ = 0.44 g/g, 88% of Y_mt_), respectively. In the case of *C. freundii*, the regulation of pH proved to be mandatory, due to lactic acid production and a subsequent drop of pH that resulted in fermentation ceasing. In the fed-batch culture of *K. oxytoca*, the BDO maximum titer reached almost 70 g/L, the Y_BDO/Gly_ and the mean productivity value (Pr_BDO_) were 0.47 g/g and 0.4 g/L/h, respectively, while no optimization was imposed. The final BDO production obtained by this wild strain (*K. oxytoca*) is among the highest in the international literature, although the bioprocess requires optimization in terms of productivity and total cost. In addition, for the first time in the literature, a strain from the species *Hafnia alvei* (*viz*., *Hafnia alvei* ACA-DC 1196) was reported as a potential BDO producer. The strains as well as the methodology proposed in this study can contribute to the development of a biorefinery that complements the manufacture of biofuels with high-value biobased chemicals.

## 1. Introduction

Environmental and climate issues, as well as the finiteness of fossil reserves, drive the need for the development of sustainable substitutes to petrochemicals and fuels. Alternative synthetic routes concerning biofuels and bulk biochemicals are currently being researched and developed. One of these is biodiesel, a renewable biofuel with lower levels of emission of greenhouse gases (e.g., CO_2_) as compared to fossil fuels. Biodiesel production, having a constantly growing industrial application and a continuously increasing production in recent years, has much to offer in the resolution of several of the above-mentioned issues [1,2]. Nevertheless, it is widely known that biodiesel production is linked with glycerol generation, whose accumulation can lead to considerable environment issues. Biodiesel-derived glycerol is the main by-product of the process; for 10 kg of biodiesel produced, 1 kg of glycerol is generated as a side-product of the process [3,4]. The constantly increasing production of biodiesel worldwide is the principal reason for the dramatical drop in its price. From a biorefining point of view, its use as a starting material for a plethora of value-added bio-chemicals holds great potential in terms of sustainability and viability of the biodiesel industry, as additional revenues can be generated. As a result, valorization of glycerol through microbial and/or chemical conversions presents enormous possibility [5]. Novel ideas and high-tech tools are important for addressing this issue, as waste valorization via microbial fermentation can significantly contribute to extending the value of every component in the chain as long as possible, complying with bioeconomy guidelines. An important contribution can be made by industrial or white biotechnology, which is related with the sustainable production of biochemicals, biomaterials and biofuels from renewable resources (e.g., glycerol), using living cells and/or their enzymes.

The metabolic pathways involved in glycerol assimilation can lead to a plethora of products (e.g., PDO, BDO, EtOH, citric acid, lactic acid, polyols, etc.), depending on the microorganism and the fermentation conditions employed [6,7,8,9,10]. Moreover, biodiesel-derived glycerol can be biotechnologically converted through a one-step process into fermentation products using microorganisms such as bacteria, fungi and yeasts [5,11], but also into even more customized added-value molecules, when chemical or enzymatic processes follow. More precisely, these products can be used as precursor molecules for the production of new bio-based chemicals and fuels. Among others, PDO is used for polymer production, such as polytrimethylene terephthalate (PTT), while BDO is converted after dehydration to methyl-ethyl ketone (MEK) and can be used as a fuel additive. There are several kinds of polymers, surfactants and flavoring agents deriving from glycerol valorization that can be produced following subsequent microbial and enzymatic or chemical processes [12,13,14,15].

The species belonging but not limited to the genera *Bacillus*, *Citrobacter*, *Clostridium*, *Enterobacter* and *Klebsiella* are among the most well-known microorganisms in glycerol conversions. Glycerol breakdown in microorganisms is divided into two main pathway branches: reductive and oxidative. In the reductive pathway, which in most cells is favored under O_2_ limited conditions, glycerol is dehydrated to 3-hydroxypropionaldehyde (3-HPA) by glycerol dehydratase; then, one molecule of NADH_2_ co-factor is oxidized to NAD^+^ for the conversion of 3-HPA to PDO by an oxidoreductase. Through the oxidative branch, glycerol can be oxidized to dihydroxyacetone phosphate (DHA-P) via two oxygen-dependent paths. Then, DHA-P is converted through an isomerase to glyceraldeyde 3-phosphate (G3P) and enters the glycolysis pathway. Pyruvic acid, the end-product of glycolysis, can be further oxidized to a plethora of metabolic products, as each one contributes to cell maintenance (ATP production, NADH regeneration, pH regulation, etc.); these include BDO, EtOH, acetoin, lactic acid and acetic acid, to name a few (Figure 1).

Several studies have revealed that although biodiesel-derived glycerol contains many impurities (e.g., methanol salts, free-fatty acids, etc.) and/or a very low pH value, it can be successfully assimilated by both prokaryotic and eukaryotic microorganisms, without any previous treatment and purification steps. A plethora of microbial compounds, including but not limited to PDO, BDO, EtOH, polyols (mostly arabitol, erythritol and mannitol), citric acid, microbial polysaccharides and microbial lipids can be synthesized through glycerol fermentation processes [11,16,17,18,19,20]. 

The aim of the present study was to investigate the capability of a relatively broad spectrum of prokaryotic microbial strains towards their potentiality to assimilate biodiesel-derived glycerol and convert it into diols, acetoin and EtOH. Newly isolated strains were tested, and microbial behavior was critically discussed for the production of these value-added molecules, in order to improve both glycerol uptake rate and product formation. The most promising of the previously screened strains were finally cultivated in both batch and fed-batch bench-type laboratory-scale bioreactors. Technological considerations of the bacterial potentialities of glycerol conversions were critically assessed. 

## 2. Materials and Methods

### 2.1. Microorganisms

The used microorganisms were as follows: *Bacillus subtilis* ACA-DC 1176, *Citrobacter freundii* FMCC-207, *Citrobacter freundii* FMCC-8, *Citrobacter freundii* FMCC-B-294, *Citrobacter freundii* NRRL-B-2645, *Enterobacter aerogenes* FMCC-9, *Enterobacter aerogenes* FMCC-10, *Enterobacter ludwigii* FMCC-204, *Klebsiella oxytoca* FMCC-197, *Klebsiella oxytoca* ACA-DC 1581, *Klebsiella* sp. EMBT-1 and *Hafnia alvei* ACA-DC 1196. Strains with the code FMCC were isolated from various foodstuffs and were identified and characterized by the Department of Food Science and Human Nutrition, Agricultural University of Athens (AUA) [21], and deposited in the institutional culture collection of this Department. The strain with the code NRRL was purchased by the NRRL Culture Collection (Peoria, IL, USA). Strains with the code ACA-DC were kindly provided by the Culture Collection of the Laboratory of Dairy Research, Department of Food Science and Human Nutrition, AUA. The code EMBT describes the strains isolated and characterized by the Laboratory of Food Microbiology and Biotechnology, Department of Food Science and Human Nutrition, AUA. Long-term storage took place at T = −80 °C in Tryptic Soy Broth, supplemented with 20% *w*/*v* glycerol. In short-term storage, strains were stored in glycerol solution (50% *v*/*v*) in freezing (T = −18 °C) and were maintained on YPDA medium (yeast extract at 10 g/L, peptone at 20 g/L, glucose at 20 g/L, agar at 20 g/L) at T = 4 °C. All microorganisms were regenerated in YPDA and were incubated for 24–48 h at T = 30 °C once per month to maintain their viability. 

### 2.2. Preculture

The preculture medium, used in order to set the microorganism in exponential phase before the inoculation of the main culture medium, was as follows (per L of deionized water): glucose 10 g, yeast extract 10 g, peptone 10 g. The preculture medium was transferred into 250-mL Erlenmeyer flasks (50 ± 1 mL working volume) and was inoculated after being sterilized in autoclave. Incubation took place in an orbital shaker (Lab-Line, IL, USA) at an incubation temperature of T = 30 ± 1 °C and a stirring rate of 180 ± 5 RPM for efficient oxygen transfer, with the fermentation time ranging from 16 to 26 h depending on the metabolic activity of each strain.

### 2.3. Principal Culture

#### 2.3.1. Main Characteristics

The fermentation medium used in all experiments contained (per L of deionized water) peptone 5 g, meat extract 5 g, yeast extract 2.5 g, K_2_HPO_4_ 2 g, CH_3_COONa 5 g, MgSO_4_·7 H_2_O 0.4 g and MnSO_4_·H_2_O 0.05 g. Pure glycerol (99.0% analytical grade, Sigma-Aldrich, Darmstadt, Germany) or crude glycerol were used as a carbon source at different initial concentrations ranging between 20 g/L and 60 g/L. Crude glycerol was derived from the biodiesel plant of Verd S.A. (Velestino, Greece), and its appearance was brownish and turbid with no visible impurities. Analyses performed by ASG Analytik-Service AG (Neusäss, Germany) indicated that the glycerol content was 74.9%, the water content was 11.8%, the ash content was 3.4%, the methanol content was 2.05% and the non-glycerol organic matter content was 9.9%. The initial pH value was set at 7.2 ± 0.1, and pH adjustment, when necessary, was performed as in Vasilakis et al. [22]. The main culture was inoculated after sterilization (121 °C, 20 min); the strains were derived from the pre-culture (ranging between 0.5 and 10% of the working fermentation volume) when the cell growth was in the exponential phase.

#### 2.3.2. Anaerobic Batch Fermentation in Duran Bottles

The fermentations were carried out in 100-mL airtight borosilicate glass (Duran) bottles (80 ± 1 mL working volume); hence, there was no input of oxygen (anaerobic conditions were therefore rapidly imposed through the self-generated anaerobiosis that occurred) during the fermentation. There was no nitrogen sparging, as strict anaerobic conditions were not required. Incubation took place in an orbital shaker (Lab-Line, IL, USA) at an incubation temperature of T = 30 ± 1 °C and a stirring rate of 70 ± 5 RPM or 100 ± 5 RPM. The inoculum size was 5% (*viz*., 4 mL)

#### 2.3.3. Aerobic Batch Fermentation in Shake Flasks

The fermentations were carried out in 250-mL flasks filled with 50 ± 1 mL working volume. Incubation took place in an orbital shaker (Lab-Line, IL, USA) at an incubation temperature of T = 30 ± 1 °C and a stirring rate of 180 ± 5 RPM. The inoculum size was 1% (*viz*., 0.5 mL).

#### 2.3.4. Anaerobic Batch Fermentation in Bioreactor

The fermentation was performed in a 3.6 L bench top bioreactor (Labfors 4, Infors HT, Bottmingen, Switzerland) with a working volume of 1.7 L at an incubation temperature of T = 30 ± 1 °C. Before inoculation of the culture, nitrogen was infused in the medium until the dissolved O_2_ reached a value lower than 0.5% *v*/*v*; then, no further gas was added. Agitation speed was 150 ± 5 RPM. The inoculum size was 10% (*viz*., 150 mL). The pH was maintained above 6.0.

#### 2.3.5. Aerobic Batch and Fed-Batch Fermentation in Bioreactor

The fermentation was performed in a 3.6 L bench top bioreactor (Labfors 4, Infors HT, Bottmingen, Switzerland) with a working volume of 1.7 L at an incubation temperature of T = 30 ± 1 °C. There was a continuous input of air (2 L/min), the agitation speed was 550 ± 5 RPM and the pH value was set by addition of NaOH (5 M) at 6, after its spontaneous fall from the initial pH = 7.1. In fed-batch fermentation, 4 pulses of concentrated crude glycerol solution occurred, when the concentration of the latter in the culture medium was above 20 g/L. The inoculum size was 10% (*viz*., 150 mL).

### 2.4. Analytical Procedures

Samples for quantitative analyses were periodically taken in order to perform kinetic and physiological studies on the microbial fermentations. The cells were harvested, as precipitate, through centrifugation at 15,000× *g*, at T = 4 °C for 10 min (Hettich Universal Centrifuge, Model 320-R, Merck KGaA, Darmstadt, Germany), and washed twice with distilled water. The supernatant was collected and analyzed by a High-Pressure Liquid Chromatography (Waters Alliance 2695, Milford, MA, USA) system equipped with an Aminex HPX-87H (Bio-Rad Laboratories, Hercules, CA, USA) column with size 30.0 cm × 7.8 mm, coupled to a differential refractometer. The mobile phase used was H_2_SO_4_ (0.005 M), with a flow rate of 0.5 mL/min (isocratic elution) and a column temperature of T = 60 °C. The samples were diluted to appropriate concentration and filtered through a 0.2 μm membrane filter before injection (injection volume = 20 μL). Metabolic products (e.g., PDO, BDO, EtOH, acetoin, lactic acid, acetic acid and formic acid) as well as substrate components (e.g., glycerol and acetic acid) were detected and subsequently quantified based on the corresponding standard curves. The total dry cell weigh (DCW, g/L) was gravimetrically determined after drying the precipitate at T = 80 °C until a constant weight was reached. 

### 2.5. Data Analysis

Each experimental point of all the kinetics presented in the tables and figures is the mean value of two independent determinations, while the standard error (SE) for most experimental points was ≤17%. 

## 3. Results and Discussion

### 3.1. Screening

In this first experimental chapter, preliminary trials were conducted in order to evaluate the assimilation level of pure and crude glycerol under different conditions as well as the variety of metabolic products by 12 prokaryotic strains. These batch cultures had an initial glycerol concentration (Gly_0_) of 20 g/L, and two different conditions were employed regarding the aeration strategy. All 12 tested strains proved to be able to assimilate both pure and crude glycerol. Some strains successfully converted glycerol into metabolic products under both aerations tested, namely the self-generated anaerobic (Duran bottles) and aerobic (shake-flasks) environments. The profile of end-products varied depending on strain and culture conditions. A plethora of metabolic products were synthetized, such as PDO, BDO, EtOH, lactic acid and acetoin. In most cases, there was no significant difference between pure and crude glycerol in terms of consumption rate and end-products.

#### 3.1.1. Self-Generated Anaerobic Cultures in Duran Bottles

The aeration regime in Duran bottles can be described as self-generated anaerobiosis; there was no nitrogen sparging, as strict anaerobic conditions were not required. Three batch cultivation conditions were carried out for 12 prokaryotic strains. More specifically, the effect of agitation speed (70 and 100 RPM) and glycerol’s purity (99.9% and 74.9%) on the microbial growth and the profile of end products were studied. The kinetic data gathered from this experimental work are presented in Table 1, in which glycerol consumption, final pH and concentrations of dry cell weight (DCW), PDO, BDO, EtOH and lactic acid are presented for each strain and each fermentation condition.

*C. freundii* FMCC-207 consumed 100% and ≈40% of the carbon source in pure and crude glycerol, respectively. At both agitation conditions tested with pure glycerol, EtOH was the predominant product, at 6.7 g/L and 7.6 g/L at 70 and 100 RPM, respectively. The production of BDO was about 3.5 g/L under both agitation conditions, while that of PDO was negligible, regardless of the agitation speed imposed. When pure glycerol was used, the pH dropped to 6.3 at the end of fermentation, which did not appear to affect the rate of glycerol uptake. On the contrary, in crude glycerol fermentation, the drop in pH value was faster and much more significant, reaching a value of 5.3 in the first 12 h of fermentation, making the environment unsustainable for further cell growth. Therefore, only 8 g/L of glycerol was consumed, with the major product being PDO (3.2 g/L). In this case, it is worth noting that implementation of different types of glycerol resulted in different pH fluctuations, quantities and percentages of glycerol consumption as well as different end products. 

*C. freundii* FMCC-8, *C. freundii* FMCC-B-294 and *C. freundii* NRRL-B-2645 were unable to consume more than 45% of glycerol under all conditions tested due to the pH drop into the medium (at a final value ≈ 5.3, no further glycerol assimilation occurred). In all cases, the dominant metabolic product was lactic acid, varying from 2.0 g/L to 3.3 g/L. Lactic acid is known to be secreted as a by-product of PDO production [23,24,25]; the enzyme responsible for lactate synthesis during glycerol metabolism is that of lactate dehydrogenase, encoded by the LdhA gene [26].

Three strains of the genus *Enterobacter* (*viz*., *E. ludwigii* FMCC-204, *E. aerogenes* FMCC-9 and *E. aerogenes* FMCC-10) were cultivated under self-generated anaerobiosis conditions. Regarding product formation in crude glycerol media, EtOH production by *E. ludwigii* FMCC-204 was 7.9 g/L, and BDO reached 3.7 g/L. This is the highest concentration of EtOH produced in all cases (see Table 1). These results revealed that glycerol, regardless its purity, was completely and rapidly consumed. PDO production was negligible despite the anaerobic conditions imposed, suggesting that the recycling of the generated NADH co-factors occurred in a metabolic network different than that of the Dha regulon (in agreement with results reported by a restricted number of bacterial strains of the genera *Enterobacter* and *Citrobacter*, screened in Metsoviti et al., 2012 [27]). Likewise, pH value was not a restricting agent for cell growth. Both strains of the genus *E. aerogenes* (FMCC-9 and FMCC-10) producing EtOH as the major product had a similar profile of metabolic products, which confirms the results of previous studies finding EtOH as the major product [28]. These strains consumed approximately 60% and 80% of pure glycerol in 70 and 100 RPM, respectively. In the latter agitation speed, the concentration of EtOH increased ≈20%, while BDO concentration remained almost the same, at ≈3.5 g/L. This could be a result of insufficient nutrient availability in the culture media due to reduced agitation speed. In crude glycerol fermentations, the concentration of end products and glycerol consumption were not significantly affected by its impurities. 

Regarding *Klebsiella* genus, 3 strains, namely *K. oxytoca* ACA-DC 1581, *K. oxytoca* FMCC-197 and *Klebsiella* sp. EMBT-1, were cultivated under self-generated anaerobiosis conditions. *K. oxytoca* ACA-DC 1581 cultured at 70 RPM partially assimilated pure glycerol, whereas in higher agitation speed (100 RPM) both pure and crude glycerol’s uptake was above 80% *w*/*w*. Concentrations of EtOH and BDO were alike and comparable in all three cultivation conditions; their maximum values appeared for the regime of crude glycerol at 100 RPM, being 4.4 g/L and 4.5 g/L, respectively. The production of lactic acid lowered the pH in the fermentation medium to an approximate value of 5.5. In all conditions tested, *K. oxytoca* FMCC-197 was able to completely assimilate glycerol (except the case of 70 RPM), produce EtOH as the major product and maintain the same profile of metabolic products. EtOH concentration reached a value of 7.5 g/L in pure glycerol at 100 RPM, while BDO and PDO were produced at a smaller scale, around 4 g/L and 2 g/L, respectively. In a previous work of Metsoviti et al., when the medium was sparged with N_2_ before autoclaving or during the fermentation, the major product was PDO [6]. *Klebsiella* sp. EMBT-1 was not able to assimilate more than 20% *w*/*w* of available glycerol under self-generated anaerobiosis conditions, although DCW was higher than 5 g/L in experiments carried out in pure glycerol, suggesting the negative effect of the impurities of the feedstock upon the bacterial metabolism. *Hafnia alvei* ACA-DC 1196 and *Bacillus subtilis* ACA-DC 1176 did not show a significant uptake of glycerol (less than 30% *w*/*w*) under any of the conditions tested. This is in contrast with the results of a previous study, when the stain *H. alvei* AD27 was able to produce PDO under anaerobic conditions [29]. 

The two main outcomes based on the results presented in Table 1 are that *E. ludwigii* stood out for its innate ability for high EtOH production and that microbial growth and further crude glycerol consumption of all *C. freundii* strains tested seemed to have been inhibited by the pH drop in the medium. At this point, it should be stated that the different final pH value between pure and crude glycerol fermentation is a result of the metabolic pathway followed by the microorganism to catabolize glycerol. For example, the fermentation of crude glycerol has been shown to favor the production of acetic acid, which lowers the pH value but also generates energy for the cell through the ATP production. This phenomenon could be an energy-producing response to overcome stress from impurities found in an unpurified substrate such as crude glycerol.

#### 3.1.2. Aerobic Culture in Shake-Flasks

The series of experimental works presented in Table 2 includes batch cultures in shake-flasks under aerobic conditions. Each bacterial strain was cultivated in pure and crude glycerol in order to study the effect of impurities on microbial growth and metabolic pathway.

In the case of the *Citrobacter freundii* strains, only *C. freundii* FMCC-207 was able to assimilate glycerol completely and only when it was in its pure form, such as in the anaerobic environment (see Table 1). *C. freundii* FMCC-207 could produce almost equal amounts of BDO (3.5 g/L) and EtOH (3.8 g/L), having a DCW of 3.5 g/L. However, in crude glycerol fermentation, its consumption was only 40%, and the main metabolic compound produced was PDO (3.3 g/L) followed by lactic acid production (2.4 g/L), which lowered the pH to 5.2, resulting in no further microbial growth. In any case, there was no EtOH production, unlike in anaerobic environment, where EtOH was the major product. The rest of the *C. freundii* strains tested were not able to assimilate more than 45% of glycerol, either when pure or crude glycerol was employed as substrate, as pH dropped to a final value of approximately 5. In most cases, PDO and lactic acid were the predominant products. 

All *Enterobacter* strains tested were able to completely consume both pure and crude glycerol. *E. ludwigii* FMCC-204, studied for the first time in glycerol assimilation, produced non-negligible amounts of EtOH, but at significantly lower levels than in anaerobic conditions. The DCW was 3- to 4-fold higher than in the anaerobic environment, suggesting that under aerobic conditions the carbon source (*viz*., glycerol) was mainly diverted to cell growth rather than to other metabolic products.

In Table 2, among the strains of the genus *Klebsiella*, *K. oxytoca* ACA-DC 1581 produced the highest concentration of BDO in both types of glycerol, namely 5.4 g/L and 5.2 g/L in pure and crude, respectively. These findings are in agreement with the results of the contemporary literature, as *K. oxytoca* is well known for its ability to produce BDO and acetoin under aerobic conditions from sugar-based substrates [30,31,32]. The major products of *Klebsiella* sp. EMBT-1 and *K. oxytoca* FMCC-197 were lactic acid and EtOH, respectively, while in both strains DCW reached approximately 9 g/L and 7.5 g/L, almost two times higher than in the anaerobic conditions. *H. alvei* ACA-DC 1196 was able to completely consume glycerol and produce a non-negligible amount of BDO (3.7 g/L), contrary to the respective results presented in Table 1. To the best of our knowledge, this is the first report that a strain from *H. alvei* species can produce BDO from glycerol fermentation, as there are no data in the literature that address BDO production by any *H. alvei* strain from any type of substrate. Finally, results from *B. subtilis* ACA-DC 1176 fermentation indicated that the glycerol catabolism was directed towards two major products, namely BDO (≈2.5 g/L) and acetoin (≈2 g/L). A recent study by Suttikul et al. focuses on BDO production by the *B. subtilis* species and the effect of dissolved oxygen [33]. At the point of the fermentation, when glycerol was completely assimilated, the ratio of BDO to acetoin was about 1.25. Nevertheless, at a later stage, the ratio was minimized to 0.1 due to the reversible bioconversion of BDO to acetoin, an observation that coincides with the recent literature [12,34]. 

Considering the above results, it is interesting that under these experimental conditions the carbon flow was mainly channeled toward biomass formation, the concentration of which significantly increased as compared to the trials performed under anaerobic conditions. Glycerol’s purity did not have a negative effect on the microbial metabolism and the level of its consumption, as crude glycerol was fully assimilated in all cases except those of *C. freundii* strains, where the pH value was a restriction agent for microbial growth. Nevertheless, *K. oxytoca* ACA-DC 1581, as reported for the first time, significantly increased its BDO production in both pure and crude glycerol.

#### 3.1.3. Trials Performed with Regulation of Culture pH

The use of crude glycerol as the sole carbon source affected the drop of pH value, which inhibited the growth of all *C. freundii* strains tested (see Table 1 and Table 2). Therefore, a new series of experiments on crude glycerol was designed, in which the pH was adjusted by adding NaOH (5 M) so that its value in the fermentation medium was not less than 5.7, in order to favor cell growth. The results presented in Table 3 demonstrate that in almost all cases, glycerol consumption, DCW and metabolic products were significantly increased due to pH adjustment. *C. freundii* FMCC-207 produced 9.3 and 7.7 g/L of EtOH in aerobic and anaerobic conditions, respectively, while without pH adjustments, EtOH production was almost zero. *C. freundii* NRRL-B-2645, which is studied for the first time, produced 6.4 g/L of PDO after pH adjustment in an anaerobic environment, which is two times higher than without pH control. 

### 3.2. Further Studies of the Most Promising Strains

#### 3.2.1. Trials with Higher Initial Glycerol Concentration

The most efficient bacterial strains were evaluated for their adaptation to higher initial concentrations of crude glycerol, in terms of cell growth and metabolic products. According to the results of Section 3.1.3, non-previously studied wild-type bacterial strains stood out for their ability to produce high value-added metabolic products, through crude glycerol valorization. Specifically, the strains *K. oxytoca* ACA-DC 1581, *E. ludwigii* FMCC-204 and *C. freundii* NRRL-B-2645 produced interesting BDO, EtOH and PDO quantities, respectively, in relatively elevated concentrations and productivities, while the conversion yields of metabolites synthesized per unit of glycerol consumed were quite high and, in some cases, close to the maximum theoretical ones. Provided that they also showed remarkable growth and noticeable crude glycerol assimilation, they were selected for further study in higher scale batch experiments. For all results presented in the next sections, crude glycerol was the sole carbon source. 

Considering that aerobic conditions led to higher productivity of BDO by *K. oxytoca*, all experiments that followed were conducted in shake-flasks. The results indicate that *K. oxytoca* was able to completely assimilate glycerol in both levels of Gly_0_ tested, namely ≈60 g/L and ≈35 g/L (Table 4). The highest yield was 0.46 g/g and occurred in the latter case, while the highest productivity of BDO (0.52 g/L/h) and glycerol consumption rate (1.3 g/L/h) occurred in the higher Gly_0_ used. These batch-type shake-flask fermentations with no optimization as regards parameters such as the agitation, feeding and oxygenation, indicate that *K. oxytoca* is a highly promising strain for BDO production [35,36]. Figure 2a,b shows the kinetics of DCW and metabolic products as a function of the time of fermentation. In both figures, acetoin production occurs at the latter stage of fermentation, after the depletion of the carbon source (*viz*., glycerol) and after the maximum value of BDO had been achieved. Under high oxygen supply conditions, NADH is oxidized to NAD^+^ through oxidative phosphorylation, leading to a limited availability of NADH, which can be regenerated via the bioconversion of BDO to acetoin. This phenomenon, which is triggered by the depletion of the carbon source and is based on the reversible reaction of acetoin to BDO, has a vital role in preventing the intracellular acidification environment, as it maintains a constant oxidation-reduction state [34].

The strain *E. ludwigii*, based on the results of the initial trials, was cultivated under self-generated anaerobiosis in Duran bottles, in which EtOH production was the highest achieved in the present study (EtOH_max_ ≈ 20 g/L; see Table 4). The assimilation of glycerol was 100% when Gly_0_ was adjusted to ≈42 g/L and 70% when it was ≈66 g/L. In addition, when Gly_0_ was ≈66 g/L, an extended lag phase occurred, resulting in a lower glycerol consumption rate (*viz*., 0.4 g/L/h vs. 0.7 g/L/h when Gly_0_ was ≈42 g/L). EtOH was the major product in both cases, providing almost an equal yield (≈0.4 g/g, which corresponds to 80% of the maximum theoretical one; see Sarris and Papanikolaou, 2016 [37]) and productivity (≈0.2 g/L/h). Although Psaki et al. studied its ability to produce BDO from the accumulation of sugarcane molasses [38], little research has been conducted regarding glycerol valorization. Data for *C. freundii* are not shown, because it was not able to perform manually pH adjustment in a higher Gly_0_; hence, a batch-culture in a bioreactor system was performed (Section 3.2.2).

#### 3.2.2. Batch Bioreactor Cultures

The experimental planning anticipated the need for further improvement of both glycerol uptake rate and product formation in batch-bioreactor experiments with pH regulation, due to the problems that were created in the previously conducted experiments (specifically with *C. freundii* strains) with pH value drop. The batch cultures had crude glycerol as the sole carbon source and were carried out in a 1.7 L working volume bioreactor under aerobic (*K. oxytoca* ACA-DC 1581) and anaerobic (*C. freundii* NRRL-B-2645) conditions. Both were selected based on their previous performance (see Table 1, Table 2, Table 3 and Table 4).

The anaerobic regime in *C. freundii* NRRL-B-2645 cultivation differs from the self-generated anaerobiosis that occurred in Duran bottle experiments (Section 3.1.1 and Section 3.1.3). Before inoculation of the culture, nitrogen was infused in the medium until the dissolved O_2_ reached a value lower than 0.5% *v*/*v*; then, no further gas was added. Agitation speed was comparatively low (150 RPM), to avoid dispersion of the remaining dissolved oxygen, and pH value was maintained above 6.9 by the addition of NaOH (10 M). Figure 3 provides data on the kinetics of microbial growth and its ability to assimilate glycerol and to produce high PDO titer.

The final PDO concentration reached 28.8 g/L at 46 h, while 51.5 g/L of the available glycerol was consumed. PDO productivity was 0.63 g/L/h, while conversion yield of PDO produced per unit of glycerol consumed (Y_PDO/Gly_) was 0.56 g/g, and the glycerol consumption rate reached 1.1 g/L/h. Y_PDO/Gly_ was equal to 93% of the theoretical maximum (≈0.60 g/g, calculated assuming a culture without hydrogen, butyric acid and other compound formations that are antagonistic to the pathway glycerol—1,3-propanediol; Zeng, 1996 [39]), while the productivity was particularly high according to the modern literature [6]. While Y_PDO/Gly_ was high, it was accompanied by a non-negligible by-product formation, as 5.2 g/L and 5.0 g/L of acetic acid and lactic acid were produced, respectively. On the other hand, it is worth mentioning that in Figure 3, there are two distinguished phases regarding the rate of glycerol consumption; in the first 19 h of fermentation, the consumption rate was 0.58 g/L/h, while from 19 h until the end (46 h), it was equal to 1.5 g/L/h, nearly threefold higher. The above may be due to a prolonged adaptation phase of the microorganism to the high glycerol concentration (≈60 g/L) and/or to the abrupt absence of O_2_ compared to the pre-culture conditions, which were fully aerobic. 

The bacterial strain *K. oxytoca* ACA-DC 1581 was cultivated under aerobic conditions, in which airflow was set at 1.5 VVM and agitation speed at 550 RPM. The initial pH was 7.0 and was left uncontrolled until it dropped to 5.8, when it was kept above 5.9 during the rest of the fermentation (see also Palaiogeorgou et al., 2019 [7]). According to Figure 4, the initial concentration of 60 g/L glycerol was completely assimilated after 32 h, giving a consumption rate of 1.9 g/L/h. The maximum BDO concentration was 19 g/L, occurring after 32 h, and its productivity was 0.6 g/L/h. In the range of 14 h and 30 h, both EtOH and DCW maintained their values, and almost all the carbon flux was navigated towards BDO synthesis, giving a specific Y_BDO/Gly_ of 0.43 g/g.

#### 3.2.3. Fed-Batch Bioreactor Cultures

Judging from the encouraging results achieved by implementation of BDO fermentation by *K. oxytoca* ACA-DC 1581 under batch cultivation, it was decided to perform a fed-batch fermentation under the same culture conditions, to evaluate the kinetic behavior of the microorganism and achieve an enhanced BDO production. During fed-batch cultivation, four pulses of sterilized crude glycerol solution were added to the fermentation medium, resulting in a total glycerol availability of 184 g/L (Figure 5). The maximum concentration of BDO and the total Y_BDO/Gly_ reached 69 g/L and 0.47 g/g, respectively, after 192 h of fermentation (Figure 6). Productivity was maximum during the first 52 h of the fermentation, that is 0.6 g/L/h, while it decreased over time and after pulses, resulting in a total Pr_BDO_ of 0.4 g/L/h. This is one of the highest BDO concentrations ever achieved with a wild strain from crude glycerol [12,40]. There was a notable increment of acetoin concentration (11.2 g/L) in the latter stages of the fermentation, following the same kinetics as in Figure 1. At 192 h of fermentation, before high levels of acetoin occurred, PDO was the main by-product (13.8 g/L). The level of dissolved O_2_ in the culture medium may affect the by-product formation. It is stated that the aeration regime of this fed-batch experiment is considered aerobic, although a more vigorous agitation speed or air flow could probably decrease PDO formation and lead to higher yield and productivity of BDO. Despite the decrement of BDO productivity, the Y_BDO/Gly_ remains stable throughout the fermentation; the correlation of BDO production and glycerol consumption is illustrated in Figure 5. Taking into consideration also the production of acetoin, the sum of BDO and acetoin production in 220 h of fermentation was ≈80 g/L, corresponding to a global yield of 0.49 g/g (=98% of the maximum theoretical yield). On the other hand, it is noted that the present bioprocess generates strong indications that *K. oxytoca* 1581 could be hampered by high BDO concentration (>69 g/L). Notwithstanding the low productivity of the fermentation, both the yield and the final titer indicate a bioprocess that deserves further optimization (e.g., aeration strategy, inoculum, nitrogen source, downstream process, etc.) in order to become an even more step-economical process [12,41,42,43,44]. 

## 4. Concluding Remarks

New findings are presented regarding novel studies of strains such as *Klebsiella oxytoca* ACA-DC 1581, *Citrobacter freundii* NRRL B-2645 and *Enterobacter ludwigii* FMCC-204 for their ability to bio-convert biodiesel-derived glycerol into bio-chemicals of great interest. The strain belonging to *E. ludwigii*, a species that has not been extensively studied in glycerol assimilation, produced ethanol with a Y_EtOH/Gly_ = 80% of the maximum theoretical yield (Ymt), without any specific optimization of the bioprocess, opening a new path for the strain in biotechnological processes. The anaerobic cultivation of *C. freundii* NRRL B-2645 resulted in a Y_PDO/Gly_ = 93% of the Ymt, only after pH value regulation. The fed-batch cultivation of *K. oxytoca* ACA-DC 1581 led to a BDO concentration of almost 70 g/L, with Y_BDO/Gly_ = 90% of Ymt; both values obtained from this wild strain are among the highest ones in the international literature. Interestingly, a strain from *H. alvei* species is reported as a potential BDO producer, for the first time in the literature. 

Understanding the feasibility and the metabolic mechanisms of these strains could favor an applicable industrial-scale bioprocess, in which biodiesel-derived glycerol will be used at the lowest purity directly for a bacterial fermentation, to produce PDO, BDO or ethanol in an economically sound process (productivity, low-cost substrate, product purification, etc.). Further work is underway to develop a holistic optimization of one of the above methods, aiming to provide the necessary financial incentive to stimulate expansion of the biorefining industry, as its by-products will be converted into high value-added products in a circular economy concept.

## Figures and Tables

**Figure 1 microorganisms-11-01424-f001:**
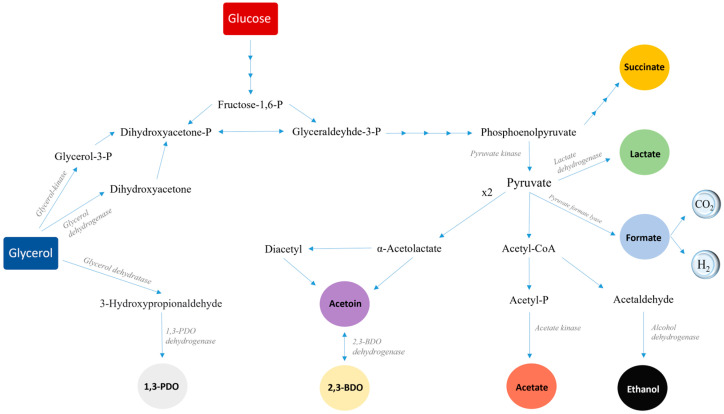
Metabolic pathways of glycerol fermentation by bacterial strains of the genera *Klebsiella*, *Citrobacter*, *Enterobacter* and *Hafnia*.

**Figure 2 microorganisms-11-01424-f002:**
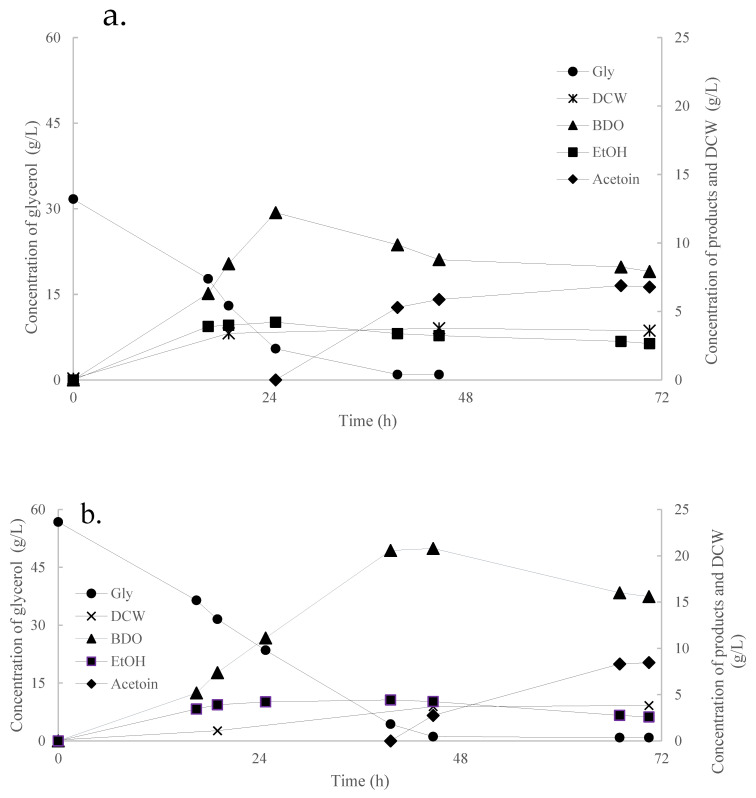
Kinetics of dry cell weight (DCW, g/L), glycerol (gly, g/L), 2,3-butanediol (BDO, g/L), ethanol (EtOH, g/L) and acetoin (g/L) evolution by *Klebsiella oxytoca* ACA-DC 1581 during batch fermentation in flasks. Initial glycerol concentration was (Gly_0_) ≈ 30 g/L (**a**) and ≈55 g/L (**b**).

**Figure 3 microorganisms-11-01424-f003:**
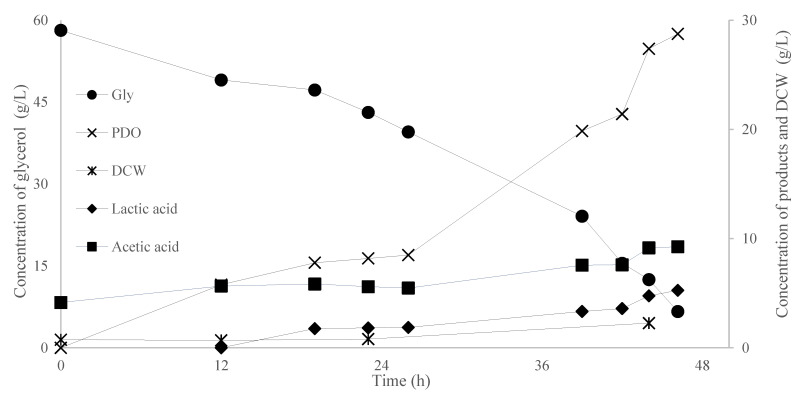
Kinetics of dry cell weight (DCW, g/L), glycerol (gly, g/L), 1,3-propanediol (PDO, g/L), lactic acid (g/L) and acetic acid (g/L) evolution by *Citrobacter freundii* NRRL-B 2645 during batch fermentation in bioreactor.

**Figure 4 microorganisms-11-01424-f004:**
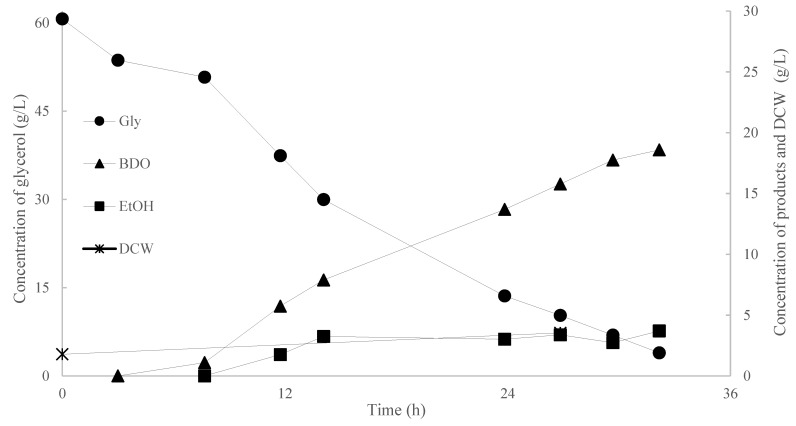
Kinetics of dry cell weight (DCW, g/L), glycerol (gly, g/L), 2,3-butanediol (BDO, g/L) and ethanol (EtOH, g/L) evolution by *Klebsiella oxytoca* ACA-DC 1581 during batch fermentation in bioreactor.

**Figure 5 microorganisms-11-01424-f005:**
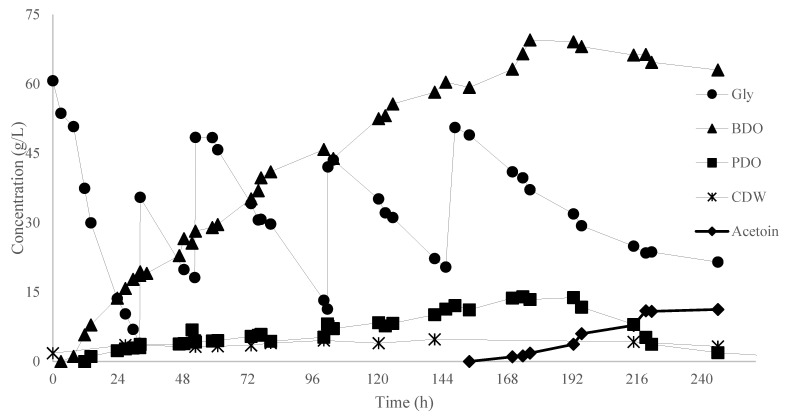
Kinetics of dry cell weight (DCW, g/L), glycerol (gly, g/L), 2,3-butanediol (BDO, g/L), 1,3-propanediol (PDO, g/L), ethanol (EtOH, g/L) and acetoin evolution by *Klebsiella oxytoca* ACA-DC 1581 during fed-batch fermentation in bioreactor.

**Figure 6 microorganisms-11-01424-f006:**
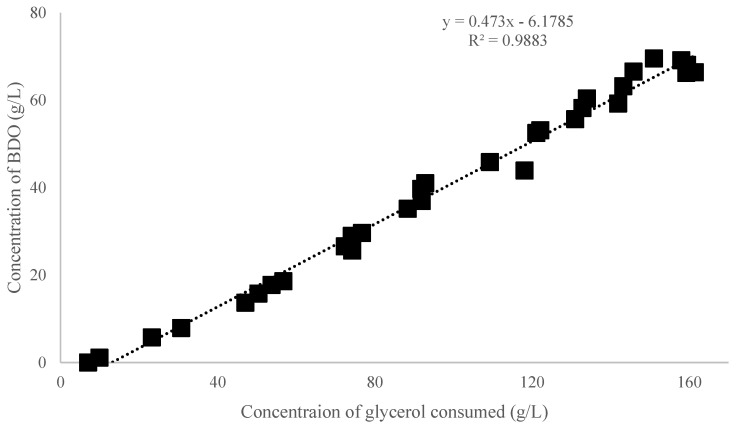
Global yield of 2,3-butanediol (BDO) production per glycerol consumption during fed-batch fermentation of *Klebsiella oxytoca* ACA-DC 1581.

**Table 1 microorganisms-11-01424-t001:** Quantitative data deriving from kinetics of bacterial strains cultivated in Duran bottles (trials under anaerobic conditions, imposed by self-generated anaerobiosis) with an initial glycerol concentration of 20 g/L; pH was not adjusted. Three different conditions were employed regarding glycerol (Gly) purity and agitation speed (RPM). Products and dry cell weight (DCW) are depicted in their maximum values, while the % of glycerol consumption (GlyC) and the pH value (pH_f_) are presented in the latter fermentation point. Only when the concentration of lactic acid (LAC) and/or formic acid (FOR) was non-negligible and higher than the concentration of a major product (e.g., BDO) is it depicted. Wp refers to weak production (<0.5 g/L). Culture conditions are as specified in Section 2. Each experimental point is the mean value of two measurements (SE < 15%).

	a. 70 RPM, Pure Gly						b. 100 RPM, Pure Gly						c. 100 RPM, Crude Gly					
Strain	PDO (g/L)	BDO (g/L)	EtOH (g/L)	DCW (g/L)	GlyC (%)	pH_f_	PDO (g/L)	BDO (g/L)	EtOH (g/L)	DCW (g/L)	GlyC (%)	pH_f_	PDO (g/L)	BDO (g/L)	EtOH (g/L)	DCW (g/L)	GlyC (%)	pH_f_
*C. freundii* FMCC-207	-	3.6	6.7	1.8	100	6.3	-	3.5	7.6	2	90	6.3	3.2	-	-	1.0	36	5.3
*C. freundii* FMCC-8	2.6	LAC 2.2	-	Wp	35	5.3	2.7	LAC 2.1	-	Wp	35	5.2	3.0	LAC 2.2	-	1.4	35	5.3
*C. freundii* FMCC-B-294	2.5	LAC 2.7	-	Wp	30	5.2	2.4	LAC 3.0	-	Wp	30	5.2	2.3	LAC 3.2	-	Wp	35	5.0
*C. freundii* NRRL-B-2645	1.4	LAC 2.0	-	Wp	20	5.4	1.8	LAC 2.3	-	Wp	23	5.3	2.7	LAC 3.3	-	Wp	45	5.1
*E. aerogenes* FMCC-9	-	3.4	5.5	1.1	65	6.0	-	3.7	6.4	1.1	82	6.2	-	3.6	6.6	1.5	90	6.5
*E. aerogenes* FMCC-10	-	3.4	4.9	1.5	61	6.1	-	3.5	6.2	1.3	80	6.4	-	3.3	5.5	1.9	85	6.6
*E. ludwigii* FMCC-204	-	4.2	6.5	3.6	100	6.3	FOR 3.3	3.8	7.6	1.5	100	5.8	FOR 3.0	3.7	7.9	1.7	100	6.1
*K. oxytoca* FMCC-197	1.6	3.7	6.0	1.7	85	6.5	1.5	4.1	7.5	2.6	100	6.3	2.2	4.0	6.6	4.0	100	6.3
*K. oxytoca* ACA-DC 1581	1.5	2.1	2.0	LAC 2.5	45	5.4	Wp	3.6	3.4	LAC 2.8	80	5.2	1.8	4.4	4.5	LAC 2.5	90	5.6
*Klebsiella* sp. EMBT-1	-	-	-	5.0	20	6.7	-	-	-	6.5	22	6.7	-	-	-	2.0	15	7.2
*H. alvei* ACA-DC 1196	-	-	1.3	1.2	26	5.9	-	-	1.6	1.4	20	6	2.9	-	2.0	1.0	30	5.9
*B. subtilis* ACA-DC 1176	-	-	-	1.1	10	6.8	-	-	-	1.0	12	6.8	-	-	-	1.2	10	7.0

**Table 2 microorganisms-11-01424-t002:** Quantitative data deriving from kinetics of bacterial strains cultivated in shake-flasks (aerobic conditions) with an initial glycerol concentration of 20 g/L; pH was not adjusted. Two different conditions were employed regarding glycerol (Gly) purity. Products and dry cell weight (DCW) are depicted in their maximum values, while the % of glycerol consumption (GlyC) and the pH value (pH_f_) are presented in the latter fermentation point. Only when the concentration of lactic acid (LAC) or acetoin (ACTN) was non-negligible and higher than the concentration of a major product (e.g., BDO) is it depicted. Wp refers to weak production (<0.5 g/L). Culture conditions are as specified in Section 2. Each experimental point is the mean value of two measurements (SE < 15%).

	a. Pure Gly						b. Crude Gly					
Strain	PDO (g/L)	BDO (g/L)	EtOH (g/L)	DCW (g/L)	GlyC (%)	pH_f_	PDO (g/L)	BDO (g/L)	EtOH (g/L)	DCW (g/L)	GlyC (%)	pH_f_
*C. freundii* FMCC-207	1.3	3.5	3.8	3.5	100	6.6	3.3	LAC 2.4	-	1.8	42	5.2
*C. freundii* FMCC-8	Wp	-	-	1.6	35	5.0	2.9	LAC 1.8	-	2.1	30	5.0
*C. freundii* FMCC-B-294	Wp	-	-	1.4	29	5.1	Wp	-	-	1.2	25	5.1
*C. freundii* NRRL-B-2645	2.3	-	1.6	1.6	30	4.8	3.1	LAC 1.5	-	1.8	45	4.9
*E. aerogenes* FMCC-9	2.5	3.6	3.4	3.5	100	8.2	1.5	3.2	3.2	3.9	100	7.9
*E. aerogenes* FMCC-10	2.3	3.8	3.4	4.0	100	8.0	1	2.6	3.2	7.6	100	7.6
*E. ludwigii* FMCC-204	1.9	2.8	-	4.3	100	6.1	1.2	2.0	3.6	4.5	100	7.7
*K. oxytoca* FMCC-197	-	3.0	4.0	7.8	100	7.7	Wp	1.8	4.7	7.2	100	7.7
*K. oxytoca* ACA-DC 1581	3.2	5.4	2.2	4.6	100	5.5	1.6	5.2	0.9	4.1	100	5.5
*Klebsiella* sp. EMBT-1	LAC 4.0	-	-	9.3	100	5.7	LAC 3.8	-	-	8.8	100	5.8
*H. alvei* ACA-DC 1196	Wp	3.7	-	4.2	100	6	-	3.1	-	4.2	100	6.2
*B. subtilis* ACA-DC 1176	ACTN 2.1	2.5	-	5.3	100	6.9	ACTN 2	2.7	-	5.0	100	7.4

**Table 3 microorganisms-11-01424-t003:** Quantitative data deriving from kinetics of bacterial strains cultivated in: Duran bottles (self-generated anaerobiosis) and flasks (aerobiosis) under controlled pH value and an initial crude glycerol concentration of 20 g/L. Products and dry cell weight (DCW) are depicted in their maximum values, while the % of glycerol consumption (GlyC) and the pH value (pH_f_) are presented in the latter fermentation point. Only when the concentration of lactic acid (LAC) was non-negligible and higher than the concentration of a major product (e.g., BDO) is it depicted. Wp refers to weak production (<0.5 g/L). Culture conditions are as specified in Section 2. Each experimental point is the mean value of two measurements (SE < 15%).

	Self-Generated Anaerobiosis						Aerobiosis					
Strain	PDO (g/L)	BDO (g/L)	EtOH (g/L)	DCW (g/L)	GlyC (%)	pH_f_	PDO (g/L)	BDO (g/L)	EtOH (g/L)	DCW (g/L)	GlyC (%)	pH_f_
*C. freundii* FMCC-207	-	3.4	7.7	4.0	100	6.5	1.0	2.2	9.3	5.7	100	7.9
*C. freundii* FMCC-8	4.3	-	-	1.5	55	5.3	1.0	-	-	2.1	50	5.2
*C. freundii* FMCC-B-294	6.0	LAC 3.5	-	1.2	100	5.4	Wp	-	Wp	2.1	55	5.7
*C. freundii* NRRL-B-2645	6.4	LAC 1.8	-	1.2	100	5.5	Wp	-	Wp	1.7	60	5.7

**Table 4 microorganisms-11-01424-t004:** Quantitative data of *E. ludwigii* FMCC-204 cultivated in Duran bottles (self-generated anaerobiosis) and *Klebsiella oxytoca* ACA-DC 1581 cultivated in flasks (aerobiosis) in elevated initial glycerol concentrations (Gly_0_); pH was not adjusted. Glycerol consumption rate (Rt_GlyC_), product titer, productivity of the major product (Pr_MajPr_), yield of the major product (Y_MajPr/Gly_) and dry cell weight (DCW) are depicted in their maximum values, while the % of glycerol consumption (GlyC) and the pH value (pH_f_) are presented in the latter fermentation point. Wp refers to weak production (<0.5 g/L). Culture conditions are as specified in Section 2. Each experimental point is the mean value of two measurements (SE < 15%).

Strain	Time (h)	Gly_0_ (g/L)	PDO (g/L)	BDO (g/L)	EtOH (g/L)	DCW (g/L)	GlyC (%)	Rt_GlyC_ (g/L/h)	pH_f_	Y_MajPr/Gly_ (g/g)	Pr_MajPr_ (g/L/h)
*E. ludwigii* FMCC-204	78	42.0	-	6.0	19.5	1.5	100	0.7	5.9	0.44	0.25
	100	66.1	-	3.6	19.5	1.8	70	0.4	5.8	0.42	0.22
*K. oxytoca* ACA-DC 1581	24.7	31.7	1.0	12.2	4.2	3.2	100	1.1	5.7	0.46	0.49
	44	57.0	3.0	20.8	4.4	3.8	100	1.3	5.7	0.39	0.52

## Data Availability

Not applicable.

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
