# Peer review of "Screening of New Industrially Important Bacterial Strains for 1,3-Propanediol, 2,3-Butanediol and Ethanol Production through Biodiesel-Derived Glycerol Fermentations"

_microorganisms, 2023, doi:10.3390/microorganisms11061424_

Round 1

Reviewer 1 Report

The purpose of the work was to investigate the capability of several microbial strains to assimilate biodieselderived glycerol and convert it into diols, acetoin and EtOH. The authors compared kinetics of the bacterial growth in different conditions (e.g. trials under areobic, anaerobic conditions, imposed by self-generated anaerobiosis, ucontrolled and controlled pH value, different RPM, pure and crude glicerol). The most promising of the screened strains, were cultivated by the authors in both batch and fed-batch bench-type laboratory-scale bioreactors. The article is an interesting piece of work and may be interesting to the broader scientific community, but it does not bring any scientifing novelty. There are literature data for aerobic and anaerobic bioconversion of crude glicerol to diols, acetoin and EtOH (including: the comparison of the the efficiency,  productivity  of the process; the analysis of the influence of the microbial cells immobilization on the efficiency of the bioconversion; the analysis of the influence of the initial concentration of crude glycerol, of medium composition or pH on the efficiency of the bioconversion during aerobic, anerobic batch, feed-batch and continuous processes etc), but the subject is important from ecological and biotechnological point of view according to circular economy concept. The experimental analysis was well designed, the content of the state of art provides useful information about the topic. As was confirmed by the authors the results of the study may be used to up-scale the described processes to industrial-scale bioprocesses, in which biodiesel-derived glycerol will be used at the lowest purity directly for the bacterial fermentation, to produce PDO, BDO or ethanol. However, the authors should revise the Materials and Methods section (Latin names of microorganisms should be italicized). Additionally, the authors provided many self-citations to the manuscript. 19 articles from 44, cited in Refernces section are the articles in which Papanikolaou S. is a co-author. Please reduce self-citations. My recommendation is to accept the article for the possible publication in “Microorganisms” after the minor revision.

Minor editing of English language is required.

Author Response

First of all, I am grateful to you for the helpful comments, as well as for appreciating our study. 

Here is a point-by-point response to your comments.

  1. Materials and Methods section (Latin names of microorganisms should be italicized).
    Thank you for pointing it out. Done.
  2. Please reduce self-citations.
    I replaced some of the self-citations [3,8,10,16].

Reviewer 2 Report

Glycerol is a by-product of the production of biodiesel, an alternative fuel to diesel. Glycerol constitutes about 10% of the biodiesel produced and its exploitation could help to reduce the production costs of the biofuel, making it more competitive with respect to liquid fuels of petroleum origin. Therefore, studies concerning the use of glycerol are of great interest, in order to obtain products with high added value. In this paper, three bacterial strains have been selected that are able to produce through glycerol fermentation processes metabolic products of interest, such as 2,3-butanediol (BDO), 1,3-propanediol (PDO) and ethanol (EtOH).

The topic covered is interesting and falls within the current concept of circular economy, which provides for less raw materials, less waste, less emissions.

The experimental design of the manuscript allows the authors to achieve the set objectives and to obtain useful data for operators in this research sector.

The bibliographic references are sufficient, even if some are very dated and the authors should replace them with more recent ones.

Overall the comments are positive. However, some modifications are required.

The captions of the tables are too long and need to be clearer and more concise. The other information can be reported in a footnote at the base of the tables and/or in the text of the manuscript.

Information about Figure 6 is missing from the text of the manuscript.

Also, as mentioned above, it is suggested to update older references.

The manuscript is easily readable and understandable in regards to the English language.

Author Response

We appreciate the time and effort that you have dedicated to providing your valuable feedback on our manuscript.

Here is a point-by-point response to your comments.

  1. The captions of the tables are too long and need to be clearer and more concise.
    Thank you for this suggestion, we tried to make the captions more compact.
  2. Information about Figure 6 is missing from the text of the manuscript.
    Thank you for pointing this out. Done
  3. Update older references.
    We have replaced the reference to Deckwer et al. with a more recent one.

Reviewer 3 Report

This Ms (Screening of new industrially important bacterial strains for 1,3-propanediol, 2,3-butanediol and ethanol production, through biodiesel-derived glycerol fermentations) is very good work, well designed and presented and informative into the field especially with the utilization of crude glycerol.

I have just minor comments to be corrected

1-Correct the formatting of scientific names at the following parts (keywords; section 2.1 at Material and methods; reference 25).

2-The crude glycerol contains 2% methanol as the authors indicated, please discuss the effect and consumption of it as much based on your analysis.

3-Section 2.4 L12; please specify the analyzed metabolic products instead of etc.

4-The tables in this study showed mixed data columns for end products and LAC/formate,,,please make columns for each product separelty,, and separate the table title from table footnotes.

Bedside the readers could hardly analyze the data in Tables 1-2  please replace with figures and add the detailed table as supplementary or try to clarify the results as better presentation and easier to the respective reader.

5-correct the formatting of reference section as the journal format

Author Response

Thank you for your valuable feedback on our manuscript.

Here is a point-by-point response to your comments and concerns.

  1. Correct the formatting of scientific names at the following parts (keywords; section 2.1 at Material and methods; reference 25).
    Thank you very much. Done 
  2. The crude glycerol contains 2% methanol as the authors indicated, please discuss the effect and consumption of it as much based on your analysis.
    It would have been interesting to explore more this aspect. However, in the case of our study, methanol concentration was significantly low in the fermentation borth (e.g., 20 g/L of Glycerol contained 0.4 g/L methanol), hence it was not quantified in HPLC analysis, as were the possible products of its catabolism (e.g., acetogenesis).

  3. Section 2.4 L12; please specify the analyzed metabolic products instead of etc.
    Aggre. Done.
  4. The tables in this study showed mixed data columns for end products and LAC/formate,,,please make columns for each product separelty,, and separate the table title from table footnotes.
    Bedside the readers could hardly analyze the data in Tables 1-2  please replace with figures and add the detailed table as supplementary or try to clarify the results as better presentation and easier to the respective reader.

    Thank you for your suggestion, however we consider that tables 1,2 should be part of the results, since the discussion of their outcomes is the main part of this study and it will not be feasible to be presented in figures. We agree that the tables will be clearer if each product had each own column, but expanding table 1 by 6 columns would make it difficult to read and fit. We could suggest to the editor to flip them horizontally, so that they are better presented.